# The Biological Clock of Liver Metabolism in Metabolic Dysfunction-Associated Steatohepatitis Progression to Hepatocellular Carcinoma

**DOI:** 10.3390/biomedicines12091961

**Published:** 2024-08-29

**Authors:** Pradeep Kumar Rajan, Utibe-Abasi S. Udoh, Robert Finley, Sandrine V. Pierre, Juan Sanabria

**Affiliations:** 1Marshall Institute for Interdisciplinary Research, Huntington, WV 25703, USA; rajan@marshall.edu (P.K.R.); udohu@marshall.edu (U.-A.S.U.); pierres@marshall.edu (S.V.P.); 2Department of Surgery, School of Medicine, Marshall University, Huntington, WV 25701, USA; finleyr@marshall.edu; 3Department of Nutrition and Metabolomic Core Facility, School of Medicine, Case Western Reserve University, Cleveland, OH 44100, USA

**Keywords:** circadian clock, liver, hepatocellular carcinoma, metabolism

## Abstract

Circadian rhythms are endogenous behavioral or physiological cycles that are driven by a daily biological clock that persists in the absence of geophysical or environmental temporal cues. Circadian rhythm-related genes code for clock proteins that rise and fall in rhythmic patterns driving biochemical signals of biological processes from metabolism to physiology and behavior. Clock proteins have a pivotal role in liver metabolism and homeostasis, and their disturbances are implicated in various liver disease processes. Encoded genes play critical roles in the initiation and progression of metabolic dysfunction-associated steatohepatitis (MASH) to hepatocellular carcinoma (HCC) and their proteins may become diagnostic markers as well as therapeutic targets. Understanding molecular and metabolic mechanisms underlying circadian rhythms will aid in therapeutic interventions and may have broader clinical applications. The present review provides an overview of the role of the liver’s circadian rhythm in metabolic processes in health and disease, emphasizing MASH progression and the oncogenic associations that lead to HCC.

## 1. Introduction

Circadian rhythms are fundamental biological systems present in most organisms that synchronize physical, mental, and behavioral changes that follow day/night cycles to harmonize various physiological events in living systems [1,2,3,4,5,6,7]. Gene mutations or behavioral changes such as irregular sleep-wake cycle, feeding/fasting behavior, and hormonal imbalance can lead to the disruption of circadian rhythm that results in metabolic dysregulation [1,2,3,4,5,6]. The genome-wide transcriptional regulation of the clock that controls these pathophysiological abnormalities operates through circadian-gene regulation [8]. The expression of the clock-controlled genes (CCGs) that function in the rate-limiting steps of various metabolic pathways is controlled by the circadian clock [9]. The details of various CCGs along with their function in circadian rhythm are depicted in Table 1. Insights into the central role of circadian rhythms in the metabolic and genomic regulation of human physiology may pave the way for future research that unravels disease mechanisms for advanced treatment strategies. Although the continuous interaction of daily oral intake, lifestyle activity, sleep patterns, and genetic background modify circadian rhythms bringing responses that vary in health and disease, we will focus on this review in the cellular interactions that circadian rhythms may exercise during the progression of metabolic disturbances in the liver associated with the metabolic syndrome.

Metabolic dysfunction-associated steatohepatitis (MASH) is a complex process that involves multiple pathogenic mechanisms such as metabolic dysfunction, chronic inflammation, immune dysregulation, fibrosis, cell cycle arrest, apoptosis, gut microbiota, epigenetic alterations, and autophagy [10,11]. These aberrant metabolic and molecular signaling pathways further complicate the liver pathology resulting in the progression of hepatocellular carcinoma (HCC). MASH associated HCC is becoming a worldwide health concern due to its increasing frequency. Of note, evidence suggests that prolonged disruptions to circadian rhythms can lead to aberrant metabolic function in liver that in turn activate various oncogenic signaling pathways resulting in the progression of HCC [12,13]. Hence, we will focus our review on the association of cellular circadian and MASH and its progression to end-stage liver disease (ESLD) and HCC.

A systematic search was conducted utilizing PubMed, ProQuest, Science Direct, and Google Scholar databases. The search was performed using the following keywords: “Circadian rhythms” OR “Biological clock” OR “Clock proteins” OR “Metabolic dysfunction-associated steatohepatitis” OR “Hepatocellular carcinoma” OR “Apoptosis” OR “Autophagy” OR “Epigenetics”. We further sorted the retrieved articles based on their relevancy. The articles without full text and published in languages other than English were excluded.

## 2. The Regulators of the Cellular Circadian/Biological Clock

Circadian homeostasis in mammals is maintained in a tissue- and organ-specific manner by a central clock located in the hypothalamus, synchronizing constantly the external solar light with our internal biological rhythms [14]. The circadian rhythm in the liver is an internal timing system that is rhythmically adapted and proactive in the response to recurring systemic and environmental changes [15]. Remarkably, any circadian clock’s genetic disturbance or environmental disruption can cause metabolic diseases or exacerbate pathological states. Complex networks of transcriptional and post-translational regulation of clock gene expression have been implicated in the homeostatic control of normal liver physiology and pathogenesis, [16,17] implying the potential involvement of the circadian clock in hepatocarcinogenesis [12].

Environmental signals perceived by the biological clock operate through the suprachiasmatic nucleus (SCN), via the retinal-hypothalamic tract (RHT) [18]. The molecular biological clock in almost all mammalian tissue types is achieved by the transcriptional translational feedback loop (TTFL) [19]. The transcriptional regulation of the core set of clock genes circadian locomotor output cycles kaput (*CLOCK*), brain and muscle Arnt-like protein-1 (*BMAL1/ARNTL*), and neuronal PAS domain protein 2 (*NPAS2*) are mediated by cryptochromes 1 and 2 (*CRY1* and *CRY2*) and the period genes (period-circadian protein homolog 1) *PER1, PER2*, and *PER3* [19]. These circadian genes heterodimerize (*CLOCK/BMAL1, NPAS2/BMAL1*) and bind to DNA sequences in the promoters of regulated genes called E-box response elements [20]. The cryptochrome and period genes act as negative feedback genes that repress the nuclear translocation of these clock gene heterodimers. Furthermore, the expression of nuclear orphan receptors *REV-ERBα* (*NR1D1*), the PAR-bZip family members, Dbp, and other clock-targeted genes are controlled by *CLOCK/BMAL1* heterodimer (Figure 1) [20,21].

## 3. Circadian Variations and Cell Differentiation

The circadian clock is known to regulate cell differentiation. Nevertheless, little is known regarding the effects of daily oscillations on cell differentiation processes. It is estimated that 10 to 15% of the transcriptome displays rhythmic expression [22], and its components are responsible for the differentiation process of various cells such as adipocytes, T cells, myoblasts, and embryonic stem cells [23,24,25,26,27]. The gradual development of circadian rhythm during developmental stages has been associated with cellular differentiation and rhythm disruptions are known to cause cancer [28,29]. Inhibiting the *BMAL1/CLOCK* pathway can lead to dysregulation of important cell cycle regulators, namely Wee1 and p21, resulting in tumor cell death. Knocking down *BMAL1/CLOCK* led to a decrease in Wee1 expression, activating apoptosis, and an increase in p21 expression, causing cell cycle arrest in the G2/M phase [30]. These findings indicate that in the liver, the circadian clock regulators *BMAL1* and *CLOCK* may play a role in promoting HCC cell proliferation by regulating the levels of Wee1 and p21, which prevent apoptosis and cell cycle arrest. The study also reveals how clock proteins maintain HCC oncogenesis and suggests cancer therapies through circadian clock modulation [12]. DNA methyltransferase1 (DNMT1) is an important regulator that helps cells differentiate by ensuring that DNA methylation occurs throughout the entire genome. When DNMT1 is lacking, the circadian clock formation that occurs during differentiation is disrupted. Significant shifts in the epigenetic environment, and consequently the transcriptome, happen during the process of embryonic stem (ES) cell differentiation, leading to the development of the circadian clock. These findings strongly suggest that epigenetic changes following pluripotency exit are necessary for the formation of a functional circadian clock [31]. In addition, malignant development is linked to abnormal differentiation or dedifferentiation processes that involve significant epigenetic changes [32,33,34,35]. These changes can include DNA hypermethylation in the promoter regions of tumor suppressor genes, like *RB1* and *CDKN2A* [36]. The loss of SMARCB1, a SWI/SNF chromatin remodeling complex component, can also lead to pediatric malignant rhabdoid tumors (MRTs) with few other genetic mutations [37,38]. These findings highlight the critical role of epigenetic dysregulation in cancer development. Interestingly, disturbances of the epigenetic landscape are also a defining characteristic of embryonic stem cell differentiation. In fact, some cancer cells share many similarities with ES cells, such as unlimited proliferation, similar metabolic needs, and epigenetic signatures [39,40,41,42].

## 4. The Association of Cell Biological Clock Regulators with Liver Metabolism

Most mammalian genes exhibit daily oscillations in expression levels, making circadian rhythms the largest known regulatory network in normal physiology that is synchronized with the external environment [43]. Circadian rhythms regulate daily bodily functions and disruption can contribute to the metabolic disturbances observed in diseases such as fatty liver and obesity [44]. Circadian regulation plays a crucial role in the metabolic pathways of the liver, especially those controlling the synthesis, mitochondrial biogenesis, oxidative metabolism, amino acid turnover, xenobiotic detoxification, and metabolism of glucose, lipids, cholesterol, and bile acid [45,46]. The rhythmic regulation of hepatic gene expression mediates the normal functioning of liver metabolism [47]. Evidence shows that circadian rhythm genes play a role in regulating liver lipid metabolism and thereby attenuate the metabolic disturbances associated with metabolic dysfunction-associated fatty liver disease (MAFLD) [48,49]. In this circadian molecular machinery, the liver’s clock regulates metabolism through circadian phosphorylation of signaling pathways involved in metabolic processes, such as mitogen-activated protein kinase (MAPK) and mammalian target of rapamycin (mTOR) pathways [50]. Disrupting the circadian clock via genetics, epigenetics, or the environment can worsen liver pathologies and trigger metabolic diseases, highlighting the clock system’s crucial role in hepatic metabolism. The daily regulation of liver transcription machinery is modulated by the coordinated action of *BMAL1* and the transcription factor C/EBPB [51]. The loss of function of the *BMAL1* gene can cause metabolic syndrome [52]. When the BMAL1 gene is deficient in mice, it increases the use of fat as energy, resulting in a higher respiratory quotient value. Fat storage in adipose tissue is reduced in the absence of *BMAL1* (*Bmal1*^−/−^ mice), with subsequent hyperlipidemia, and liver/skeletal muscle steatosis. Interestingly, the mice that lack *BMAL1* only in the liver or skeletal muscle did not develop ectopic fat formation even under high-fat diet conditions [53,54]. The deletion of BMAL1 in the mice liver leads to reactive oxygen species (ROS) accumulation, mRNA methylation, and downregulation of peroxisome proliferator-activated receptors gamma (*PPAR-γ*) transcription, affecting β-lipid oxidation [55]. Heme, a cofactor in cellular metabolism, is the natural ligand for *REV-ERBα*, a nuclear receptor that regulates circadian rhythm and metabolism. Heme binding to *REV-ERBα* inhibits the expression of genes involved in liver gluconeogenesis [56]. *REV-ERBα/β* double knockout mice exhibited higher levels of serum glucose and triglycerides compared to wild type, along with fragmented locomotor activity and reduced circadian rhythm [57]. Reports suggest that depletion of *REV-ERBα* and *REV-ERBβ* has severe effects on the cell-autonomous clock and hepatic steatosis [58].

Non-parenchymal cells may affect the liver’s circadian clock. Neutrophils activate hepatic c-Jun N-terminal kinases (JNKs) influencing the liver circadian clock by the transcription of BMAL1 and the repression of the cycling hepatokine fibroblast growth factor 21 (FGF21) [59]. These are events that impact liver lipid metabolism by activating lipogenesis and reducing FGF21’s metabolic regulatory effect. Thus, hepatic JNKs act as circadian activators in the mouse, [59] and its disruption results in a paucity of bile acid production with the development of cholangiocarcinoma [60]. Indeed, post-transcriptional mechanisms also play a key role in regulating liver functions, where bile acids are under circadian regulation harmonizing periods of feeding and fasting [61,62,63,64]. Cholesterol 7α-hydroxylase (CYP7A1), a key enzyme in hepatic bile acid biosynthesis shows an mRNA rhythm expression in rodents and enzyme activity in human serum [65,66,67]. Likewise, cholesterol synthesis exhibits a rhythm in rodents that is synchronized with the timing of food intake, [68] while in humans, cholesterol synthesis has shown no rhythm [69]. Maximizing nutrient utilization and storage relies on precise regulation of hepatic metabolism through circadian-controlled hepatobiliary pathways. Duez et al. reported that the clock gene *REV-ERBα* regulates *CYP7A1* gene expression, possibly by binding to the small heterodimer partner (SHP) promoter to prevent transcription and resulting in the de-repression of *CYP7A1* by SHP in a time-dependent manner [70]. *PER1/2* double knockout mice showed elevated levels of bile acids in their serum and liver, as well as increased levels of hepatic enzymes in their serum, indicating liver damage [71].

The peripheral clocks play a major role in regulating feeding rhythm by modulating the metabolic feedback signals of the brain, and liver plays a critical part in these metabolic processes. Many metabolites exhibit oscillations in the tissues and plasma, and evidence suggests that disruption of peripheral clock expression contributes to the development of systemic metabolic dysfunction. Hormones like glucagon and insulin, help to regulate liver glucose metabolic pathways like glucose uptake, gluconeogenesis, and glycogenolysis to ensure the proper functioning of the body, and daily plasma rhythms in these hormones have been identified in rodents and humans [72,73]. The dysregulation of glucose metabolic pathways results in type 2 diabetes and insulin resistance that further disrupts the glucose homeostasis. The circadian rhythm of plasma glucose levels can be evidenced in both rodents and humans by its peak concentration near the onset of activity [74]. Evidence suggests that CNS-lesioned rats failed to exhibit the circadian rhythmic peak of glucose levels, driven by a 24 h rhythm in insulin sensitivity responsive to either ad libitum or scheduled feeding, highlighting the pivotal role of CNS in this process [74,75]. Mice lacking a functional pancreatic clock had impaired insulin release and glucose intolerance, indicating circadian control of insulin is crucial for maintaining glucose homeostasis during feeding and fasting periods [76].

The circadian clock regulates plasma and tissue lipids, including triglycerides, cholesterol, and free fatty acids. After meals, triglycerides are delivered to the liver where they are stored or oxidized. During fasting, fatty acids from lipolysis on adipose tissue are also delivered to the liver. High-fat diets can affect peripheral clock function suppressing gene expression in mice [77]. Leptin is a circulating hormone secreted by white adipose tissue, which displays a circadian rhythm. It controls hunger and resting energy use is regulated by leptin binding to receptors in the hypothalamus, liver, and other organs. Leptin inhibits insulin secretion in pancreatic islets and decreases lipid-lowering effects in obese rats [78]. Mice with a mutation in the leptin gene (ob/ob) failed to receive proper signals to the brain, leading to hyperphagia, hyperglycemia, and hyperinsulinemia. In addition, ob/ob mice have abnormal peripheral clock gene expression compared to wild-type mice [79,80,81]. Data from non-diabetic individuals shows that fasting glycemia can be associated with four *CRY2* SNPs, whereas carriers of these SNPs’ minor alleles are involved in elevated fasting glycemia and reduced liver fat content. The correlation of *CRY2* mRNA expression with hepatic triglyceride and glucose metabolism in human tissues highlights the pivotal role of *CRY2* in hepatic fuel metabolism [82]. Timing and frequency of feeding affect liver’s triglycerides storage in the wild-type mice, as nighttime-restricted feeding decreased hepatic triglycerides by 50% [63]. Additionally, triglyceride oscillations can occur despite the absence of a functional circadian clock [63].

## 5. Cell Circadian Clock Regulators and Cell Apoptosis

The circadian clock controls the physiological functions of various tissues and organs through molecular pathways. It regulates cell growth, DNA repair, angiogenesis, apoptosis, metabolism, immune response, and inflammation [83,84,85]. Recent evidence suggests that there is a link between the biological clock and the cell cycle, with core clock proteins and key cell-cycle regulators playing a crucial role in controlling further apoptosis [12,86]. The participation of circadian rhythms in the process of apoptosis can be evidenced by the mutation of *CRY* that can activate p53-independent apoptosis pathways followed by the elimination of pre-malignant and malignant cells to delay tumor progression [87].

A *CRY* mutation can sensitize p53 mutant and onco-genetically transformed cells to TNFα-initiated apoptosis by interacting with the NF-κB pathway. This signal reduces pro-survival NF-κB and explains the delayed onset of tumorigenesis in clock-disrupted p53 mutant mice. The findings suggest unique therapeutic strategies for p53 mutation-associated cancers [87]. Interestingly, the inhibition of the p53 pathways can be correlated to diethyl nitrosamine-induced apoptosis resistance in hepatocytes of the *CLOCK* gene mutant mice [88]. In addition and in both in-vitro and in-vivo models of HCC, *NPAS2*-mediated inhibition of CDK2/4/6 and B-cell lymphoma 2 (BCL-2) phosphorylation by upregulating CDC25A has been reported to promote cell proliferation by inhibit mitochondrial apoptosis [89]. Evidence suggests that the interference of the *CLOCK* gene is associated with the downregulation of BAD and BIM proteins that in turn lead to the suppression of mitochondrial apoptosis pathways in hepatocytes [90]. In addition, the same group has also shown that expression of mitochondrial apoptosis factors such as apoptosis-inducing factor (AIF), cytochrome C somatic (CYCS), apoptotic protease activating factor-1 (APAF-1), and second mitochondria-derived activator of caspase (SMAC) was also downregulated through the interference of the *CLOCK* gene. This will ultimately result in the blockage of the apoptosome formation and the process of DNA degradation to further inhibit apoptosis process. The proteins *BMAL1* and *CLOCK* that regulate the circadian clock play a significant role in promoting the proliferation of HCC cells. They influence the levels of Wee1 and p21, which in turn prevent apoptosis and cell cycle arrest, ultimately helping to maintain HCC oncogenesis [12]. Alterations of the p21 phase due to mutation may impact cell phase locking and trigger cell cycle progression, which could be relevant for cancer pathogenesis. A mutation in the *PER2* gene has been shown to increase resistance to apoptosis and oncogenic transformation and affect tumorigenesis in mice with a cancer-sensitized genetic background [91]. These findings provide not only further understanding of the impact of clock proteins on cells and offer a potential blueprint for developing cancer therapies targeting the circadian clock, but they may unveil the probable pathogenesis of circadian rhythm-mediated apoptosis in the regulation of liver pathology.

## 6. Cell Circadian Clock Regulators and Cell Autophagy

Autophagy is a natural process that helps maintain nutrient and cellular balance. Several factors like diet, drugs, and aging can influence autophagy [92]. Cell autophagy also follows a clock-dependent pattern, activating rhythmically regardless of the organism’s health status or disease [93,94,95,96]. In turn, autophagy can impact the circadian rhythm by breaking down circadian proteins [97,98]. Autophagy-related genes and pathways exhibit diurnal rhythmicity [99]. CCAAT/enhancer-binding protein β (C/EBPβ) plays a critical role in regulating the expression of certain genes that are involved in autophagy, such as Unc-51-like kinase 1 (ULK1), LC3B, BCL-2 interacting protein 3 (BNIP3), and GABA Type A Receptor Associated Protein Like 1 (GABARAPL1). These genes exhibit oscillating expression patterns that are influenced by both circadian and nutritional signals [100]. Ma et al. found that knocking down C/EBPβ via adenoviral-mediated RNAi eliminated circadian-regulated autophagy. The same authors also discovered that liver-specific *BMAL1* knockout mice exhibited altered C/EBPβ levels, disrupted rhythmic regulation of autophagy, and decreased autophagic gene expression [101]. Interestingly, low autophagic flux occurred during feeding in the dark phase, while 24 h starvation did not affect the rhythmicity of autophagy gene expression, suggesting that the cyclic regulation of autophagy is strongly influenced by nutritional signals.

Peroxisome proliferator-activated receptor-γ coactivators (*PGC-1α* and *PGC-1β*) can activate mitochondrial gene programs in various cell types, using genes from both nuclear and mitochondrial genomes [102]. The expression of these genes also follows a strong circadian rhythm in the liver and skeletal muscle. PGC-1α may regulate mitochondrial turnover in a circadian rhythm, according to its association with circadian gene expression [103,104]. Mice without *PGC-1α* have irregular light/dark cycles [104], indicating the potential link between *PGC-1α* and the circadian system and their connection to autophagy. *PGC-1β* has been found to be a C/EBPβ transcriptional coactivator by direct bounding in 293T cells [105]. Results showed that C/EBPβ-mediated regulation of some important autophagy genes in hepatocytes has been shown to be promoted by the overexpression of *PGC-1β*, emphasizing the co-activator action of *PGC-1β* on C/EBPβ, which promotes autophagy [105]. Taken together, these findings suggests that the circadian signaling of cell autophagy can be regulated by the rhythmic activation of *PGC-1β* and *C/EBPβ*.

TFEB and TFE3 are important transcription factors that are known to activate various genes associated with autophagy, lysosomal biogenesis, and lysosomal exocytosis [106,107]. The clock-mediated autophagy gene expression can be promoted by a key clock machinery component, *REV-ERBα* (*NR1D1*), in association with the rhythmic activation of these transcription factors [108]. In addition, the activation of rhythmic expression of genes involved in autophagy can also be evidenced by the common promotor regions of *TFEB/TFE3* and *REV-ERBα*, which provides new insight into the circadian balance of these transcription factors in cell autophagy [108].

## 7. Epigenetic and Posttranslational Regulation of Liver Circadian Rhythms

Extensive epigenetic histone modifications, such as methylation and demethylation, occur with circadian-regulated gene expression in various stages of disease progression [109,110,111]. Whole-genome sequencing analysis of HCCs identified the influence of etiological background on somatic mutation patterns in multiple chromatin regulators such as AT-rich interactive domain-containing proteins (ARID1A, ARID1B, ARID2) and mixed lineage leukemias (MLL and MLL3) related to the pathogenesis of HCC [112]. Histone-remodeling enzyme MLL3 as a clock-controlled factor can regulate many epigenetically targeted circadian output genes in the liver. The rhythmic activation of core clock genes promoters such as *BMAL1*, *mCRY1*, *mPER2*, and *REV-ERBα* can be affected by the catalytic inactivation of the histone methyltransferase activity of MLL3. These findings emphasize the importance of rhythmic histone methylation in the genome-wide control of transcription [113]. MacroH2A1, a histone variant of H2A, act as a transcriptional modulator, involved in the pathogenesis by modulating the expression of oncogenes and/or tumor suppressor genes [114,115,116]. The loss in HCC cells of macroH2A1 induces the up-regulation of *PER1*, modulating the expression of circadian genes in the setting of MASH-associated HCC [116].

Sirtuin-1 (SIRT-1) is an NAD^+^-dependent deacetylase that acts as a transcription factor for many different physiological processes, is implicated in alcohol-induced liver injury, and is overexpressed in human HCC [117,118]. SIRT1 binds *CLOCK-BMAL1*, promoting the deacetylation and degradation of *PER2* and thereby repressing transcription [119]. NCOR1, nuclear receptor corepressor-1, functions as an activating subunit for the chromatin-modifying enzyme histone deacetylase-3 (Hdac3). The dysregulation of clock genes has been observed in mice with disrupted NCOR1-HDAC3 interaction along with abnormal circadian behavior by *BMAL1* expression [120]. The co-localization of the circadian nuclear receptor *REV-ERBα* with HDAC3 has been shown to regulate lipid metabolism which is further evidenced by the occurrence of hepatic steatosis in mice with selective deletion of either HDAC3 or *REV-ERBα* in the liver [121]. Hence, it can be concluded that the circadian regulation of normal hepatic lipid homeostasis is guided by the genomic recruitment of HDAC3 by *REV-ERBα*.

DNA methylation is one of the major epigenetic modifications that is involved in the transcriptional regulation of genes and conserving genome stability. Various cancers have a special epigenetic alteration that is commonly characterized by aberrant DNA methylation [122]. There is a decreased expression of *PER1*, *PER2*, *PER3*, *CRY2*, and *TIM* in HCC [123]. The decreased genetic expression was not because of mutations, but by a promoter methylation enhancing the role of epigenetic modification of circadian genes in HCC. The downregulation of these circadian genes may result in a disturbance of the cell cycle, which is correlated with tumor progression [123]. The promoter methylation in HCC cells was also observed in *PER1* and *CRY1* [124]. N6-methyladenosine methylation, one of the most prominent epigenetic modifications, has been involved in the circadian rhythm regulation of the transcripts from *PER1*, *PER2*, *PER3*, *CLOCK*, *NR1D1*, and *NR1D2* [125]. The expression of ALKBH5, the enzyme involved in N6-Methyladenosine methylation, was downregulated in HCC and pancreatic cancer. It correlated with *PER1* expression and poor patient survival [126,127,128]. In HCC patients, circadian period elongation and RNA processing delay can be induced by the knockdown of METTL3, one of the main N6-Methyladenosine methylation-associated factors, which leads to poor cancer prognosis [129]. In summary, the regulation of circadian transcription is modulated by various epigenetic factors and epigenetic mechanisms, which are vital mediators of environmental variations modulating rhythmic gene expression [111,130]. Gene expression in HCC, is commonly due to epigenetic alterations, such as DNA methylation, histone modification, and miRNA-mediated processes, critically associated with various mechanisms of proliferation and metastasis [131,132].

## 8. The Role of Circadian Rhythms in HCC Progression

HCC, the most common lethal form of liver cancer, which accounts for more than 80% of liver cancers, is one of the leading causes of cancer-related deaths worldwide [133]. The steady increase in the incidence of HCC can be evidenced by the predicted global increase of liver cancer by 55% between 2020 and 2040 [134]. Furthermore, an increase of 56.4% in the death rate compared to 2020, with approximately 1.3 million people affected worldwide, is predicted. The latest report from the North American Association of Central Cancer Registries estimates that the number of new HCC cases diagnosed in the USA is 41,210, which represents a threefold increase in incidence over the past four decades [135,136]. Evidence also underscores a higher incidence and projected increase of MASH-related HCC compared to virus-associated HCC by 2030 [137,138]. HCC is characterized by silent and slow tumor growth, where clinical diagnosis is often made in the advanced stages. Liver cancer, a major public health concern globally, is closely linked to out-of-balance circadian cues. Therefore, it is imperative to elucidate the molecular mechanisms and identify novel therapeutic targets for the successful implementation of therapies with an intent to cure strategy. The circadian clock maintains the physiological homeostasis of the liver and influences hepatocarcinogenesis. The circadian rhythm plays a crucial role in liver homeostasis by regulating the expression of many protein-coding genes through internal molecular mechanisms [139]. As circadian clock genes regulate the cell cycle checkpoint determination, genomic stability, and DNA repair, an alteration in the circadian coordination of liver metabolism can elicit the progression of liver tumorigenesis in MASH.

Studies have demonstrated the anomalous circadian function in HCC through the differential expression of several core clock genes across time [140,141]. The sympathetic nervous system pathways of circadian rhythm that target peripheral organs through adrenergic receptor-mediated intracellular signaling relate to metabolic disorder, oncogenic activation, neoplastic growth, and tumor initiation [142,143,144,145]. *BMAL1* and *CLOCK* regulate HCC cell proliferation by controlling Wee1 and p21 levels, preventing apoptosis and cell cycle arrest [12]. Chronic jet lag can cause spontaneous development of HCC in wild-type mice, and it has a role in obese humans that initiates and/or enhances the development of MAFLD to steatohepatitis (MASH) and fibrosis (ESLD) [146]. Furthermore, ablation of the farnesoid X receptor (FXR) leads to a significant increase in enterohepatic bile acid levels and jet-lag-induced HCC. Conversely, the loss of constitutive androstane receptor (CAR), a well-known liver tumor promoter that mediates toxic bile acid signaling, inhibited MASH-induced hepatocarcinogenesis [146]. Prolonged disruption of the internal clock is enough to cause spontaneous hepatocarcinogenesis by causing persistent liver metabolic dysfunction and activating oncogenes. The significance of circadian rhythms in hepatocellular carcinoma and their role in clinical settings is schematically depicted (Figure 2).

Research into the molecular mechanisms by which disruptions in the circadian rhythm drive HCC development is still in its nascent stage. In mice, deletion of germline or tissue-specific core circadian genes can promote genomic instability and hasten the development of tumors and cancer [147]. It has been shown in mice, that persistent jet lag after exposure to the carcinogen diethyl nitrosamine (DEN) increases the incidence of tumor growth by up-regulating the expression of the oncogene *c-MYC*, and by down-regulating the expression of the tumor suppressor *TP53* [148]. In mice, DEN was also reported to disrupt circadian rhythms of rest activity and body temperature [148,149]. Additionally, other studies have reported that disruption of circadian rhythm leads to over-expression of the constitutive CAR, which promotes liver tumors via the pathogenic progression of MASH to HCC [146]. The steroid receptor coactivator-2 (*SRC-2*) is another important regulator of circadian behavior directed by the dark and light cycle to regulate the metabolic homeostasis of the liver. Chronic circadian disruption in SRC-2 knockout mice exhibits a tumor-promoting activity. Of note, *SRC-2* knockout mice demonstrated an inability to adjust to the stress of long-term circadian disturbance that stimulates MASH and HCC gene signatures [150]. *NPAS2*, a core circadian molecule was frequently upregulated in HCC, promoting cell survival by enhancing proliferation and inhibiting apoptosis, leading to poor prognosis. Mechanistically, its role is mediated by transcriptional upregulation of CDC25A phosphatase, leading to dephosphorylation of CDK2/4/6 and BCL-2, inducing proliferation and inhibiting apoptosis in HCC. Zhao et al. proposed that single nucleotide polymorphisms (SNPs) in circadian genes could be potential prognostic biomarkers [151]. In a cohort of 337 unresectable HCC patients, the effect of 12 functional SNPs in five circadian genes (*CRY1*, *CRY2*, *PER1*, *PER2*, and *PER3*) was evaluated. SNP rs2640908 was found to be predominant among late-stage HCC patients, especially those who were elderly, with large tumor size, increased serum α-fetoprotein (AFP) levels, and advanced tumor node metastasis (TNM) stage [151]. Hence, the molecular insight into the critical regulatory interactions between clocks and cellular metabolism that trigger HCC progression may lead to improved disease treatment and prevention.

## 9. Circadian Genes and Therapeutic Targets in HCC

Strategies for HCC therapy developed under a multidisciplinary approach have considerably improved patients’ overall survival during the past two decades [152,153,154]. The therapeutic options include surgical treatments such as tumor resection and liver transplantation, loco-regional interventions such as ethanol administration and thermal ablation (Radiofrequency ablation-RFA or microwave ablation-MWA), transarterial interventions (Trans arterial Chemoembolization-TACE, yttrium-90), radiation (Stereotactic body radiation therapy-SBRT), and targeted therapies (gene and immune) [155]. In addition, in MASH, there is evidence that also demonstrates the association of bariatric surgery with a decreased risk of HCC [156]. Strategies for early diagnosis of disease stages and HCC prevention after successful therapeutic interventions are crucial to achieve a better overall survival rate. The cross-disciplinary knowledge of the natural history, developmental stages, and complex biological mechanisms of HCC has improved the understanding of tumor biology that facilitated the clinically oriented stratification of patients and the advanced treatment modalities to optimize survival benefits [153,154]. Recently, gene expression profiling and proteomic analyses further explored the crucial molecular events underlying HCC progression and paved the way for future research to identify novel diagnostic and therapeutic targets.

The biological clock is driven by a transcription-translation feedback loop including various transcription factors, nuclear receptors, and regulators of DNA binding, among other factors [157]. Genome-wide dysregulation of clock genes has been reported in various human cancers which can be correlated to the different stages and aggressiveness of cancer progression [158]. Dysregulation of the circadian rhythm is associated with the development and poor prognosis of cancer and circadian timing of anticancer medications has improved treatment tolerability and efficacy [159,160]. A prediction study showed a 5-fold higher survival rate in metastatic colorectal cancer patients with normal circadian rhythms compared to those with severely disrupted circadian rhythms [161,162]. A mathematical model of transcription profiles of clock genes presents the optimal timing for improved efficiency of chemotherapy [159]. The systemic and organ-specific toxicities of several anticancer medications have shown variability in their effectiveness in accordance with the fluctuations in circadian timing [160]. Hence, advances in cancer chronotherapy with optimal treatment time in the circadian rhythm may maximize anti-tumor effects and lower off-target effects. An integrative analysis of 14 CRD genes in liver cancer cells using a single-cell transcriptomic dataset demonstrated the prognostic utility of circadian rhythm disruption in various stages of tumor progression and survival [163]. Malignant cells with high CRD scores were enriched in specific metabolic pathways and were significantly correlated with macrophage infiltration. Another study identified four circadian rhythm-related genes, viz. *CRY2*, casein kinase 1 delta (CSNK1D), f-box and leucine-rich repeat protein 21 (FBXL21), and *PER1*, for the prognosis of HCC and the prediction of immune infiltration sensitivity [164]. A prognostic risk model for HCC was established using 11 circadian clock-related long noncoding RNAs (lncRNAs) and the risk score analysis showed a significant association with tumor immunity that can guide more effective prognostic and therapeutic strategies [165]. This is ground-breaking in understanding the possible prognostic and therapeutic applications of the circadian clock in the growth and progression of HCC tumors. This understanding may help drug pharmacokinetics and pharmacodynamics that are strongly influenced by rhythmic oscillations, which impact drug absorption, distribution, metabolism, excretion, and expression of drug targets in HCC.

## 10. Conclusions

The present review highlights the importance of circadian rhythm regulation in the metabolic disturbances governing the progression of MASH to HCC. Molecular clock disruption has been implicated in the pathogenesis of liver cancer and therapy resistance. Circadian rhythm has a pivotal role in liver metabolism, epigenetic changes, cell differentiation, apoptosis, and autophagy. The overall summary of the multifactorial pathophysiological alterations mediated through the disruption of circadian rhythm that favors oncogenic pathways and chromatin remodeling in HCC is depicted in Figure 3. In this sense, molecular constituents of the circadian clock may become prognostic predictors of disease severity, serving as a conduit for the development of novel therapeutic targets that explore the clinical utility of personalized therapy in liver cancer progression.

## Figures and Tables

**Figure 1 biomedicines-12-01961-f001:**
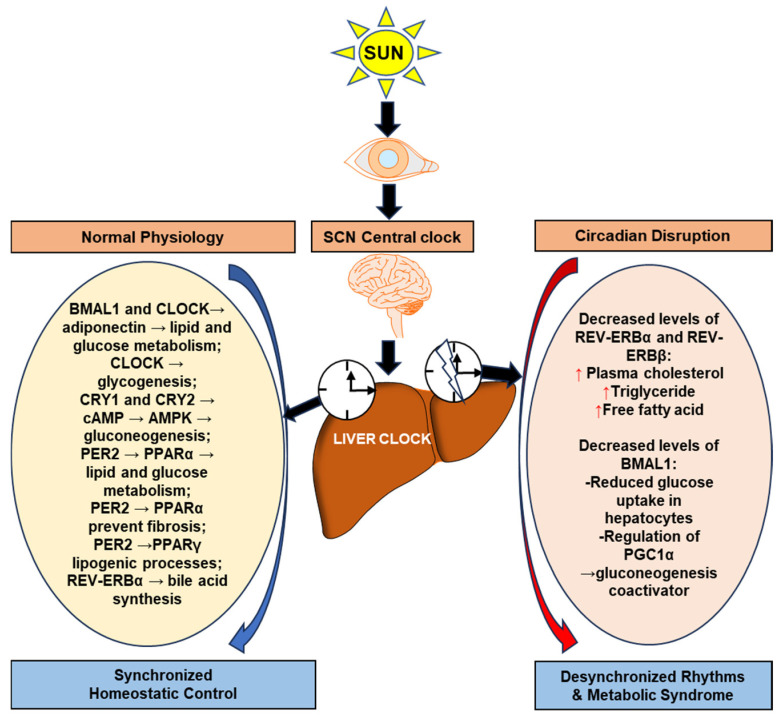
Role of circadian rhythm genes in liver metabolism. The activation of clock genes in cellular biology controls the daily fluctuations in normal liver physiology. The SCN generates internal biological rhythms, ensuring that the body’s internal processes are in synchronized with the external environment. The synchronized homeostatic control in the liver, which operates through various clock genes, modulates the expression profiles of major genes involved in the key metabolic pathways of the liver including glucose and lipid metabolism, gluconeogenesis, lipogenesis, fibrotic progression, bile acid synthesis, etc. Disruption of circadian rhythms can lead to disturbances in these natural rhythms leading to a wide range of metabolic syndromes and liver pathologies. Abbreviations: suprachiasmatic nucleus (SCN); circadian locomotor output cycles kaput (*CLOCK*); brain and muscle Arnt-like protein-1 (*BMAL1*); nuclear receptor subfamily 1, group D, member 1 (*REV-ERBα/REV-ERBβ*); period-circadian protein homolog 2 (*PER2*); cryptochrome 1 (*CRY1*); cryptochrome 2 (*CRY2*); peroxisome proliferator-activated receptors alpha (PPARα); peroxisome proliferator-activated receptors gamma (*PPARγ*); adenosine monophosphate-activated protein kinase (AMPK); cyclic adenosine monophosphate (*cAMP*); Peroxisome proliferator activated receptor gamma coactivator1 alpha (*PGC1α*).

**Figure 2 biomedicines-12-01961-f002:**
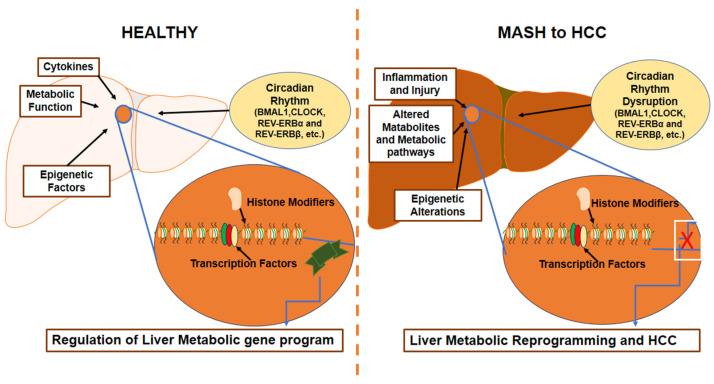
The significance of circadian rhythms in HCC. Complex gene regulatory networks, coordinated by various factors including circadian rhythm, circulating cytokines, metabolites, and epigenetic factors play a key role in liver homeostasis. Disruption in circadian rhythm along with inflammation, injury, metabolic dysfunction, and epigenetic alterations can lead to HCC development characterized by the dysregulation of the liver metabolic gene network that reflects the reprogramming of the liver. Abbreviations: hepatocellular carcinoma (HCC); metabolic dysfunction-associated steatohepatitis (MASH); circadian locomotor output cycles kaput (*CLOCK*); brain and muscle Arnt-like protein-1 (*BMAL1*); nuclear receptor subfamily 1, group D, member 1 (*REV-ERBα/REV-ERBβ*).

**Figure 3 biomedicines-12-01961-f003:**
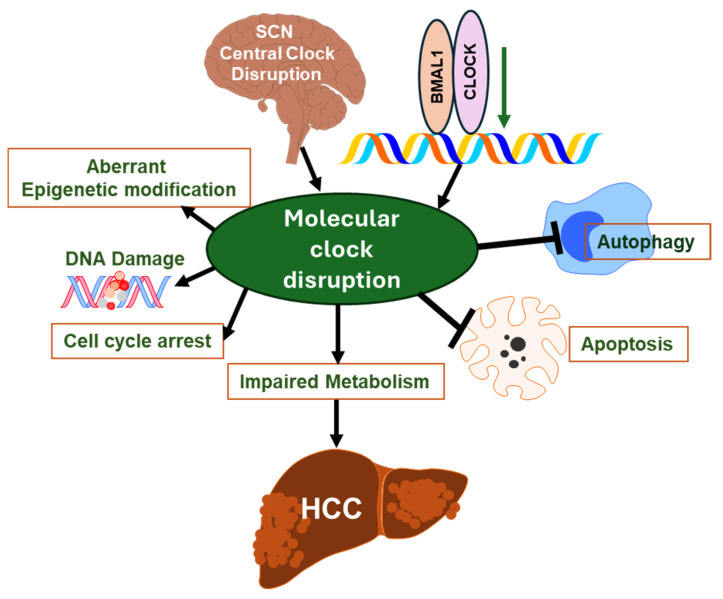
The multifactorial alterations that influence disruptions of the circadian rhythms. The central clock disruption and dysregulation in clock genes affect the molecular circadian rhythm in the liver. The disrupted circadian rhythm in the liver leads to aberrant epigenetic modification, DNA damage that causes cell cycle arrest, impaired metabolic processes, and blockage in both autophagic and apoptotic processes that eventually progress to HCC. Also, these disruptions in HCC favor oncogenic pathways and chromatin remodeling, further exacerbating the disease pathology. Abbreviations: suprachiasmatic nucleus (SCN); circadian locomotor output cycles kaput (*CLOCK*); brain and muscle Arnt-like protein-1 (*BMAL1*); hepatocellular carcinoma (HCC).

**Table 1 biomedicines-12-01961-t001:** The mammalian circadian genes.

Circadian Genes	Circadian Function
*CLOCK*	bHLH-PAS domain containing transcription factor, Positive Regulator, co-activator of PERs-CRYs transcription; Histone Acetyltransferase
*BMAL1*	bHLH-PAS domain containing transcription factor, Positive Regulator
*REV-ERBα*	Nuclear Receptor, Negative Regulator; Repressor of *BMAL1* transcription and regulator of clock-controlled genes
*REV-ERBβ*	Paralog of *NR1D1*
*NOCT*	mRNA Deadenylase
*PGC-1α*	Transcriptional coactivator
*PER1*	Co-repressor of *CLOCK-BMAL1*; PAS-domain containing negative regulator
*PER2*	Co-repressor of *CLOCK-BMAL1*; PAS-domain containing negative regulator
*PER3*	Influence chronotype
*CRY1*	Negative Regulator/Co-repressor of *CLOCK-BMAL1*
*CRY2*	Negative Regulator/Co-repressor of *CLOCK-BMAL1*
*ARNTL*	Circadian clock-regulated transcription factor promoting expression of genes involved in angiogenesis and tumor progression; Dimerization partner of *CLOCK/NPAS2*; co-activator of *PERs-CRYs* transcription
*TIM*	Role in the production of electrical oscillations output of the SCN
*ROR*	Nuclear Receptor, Positive Regulator
*RORα*	Activator of *BMAL1* transcription and regulator of clock-controlled genes
*RORβ*	Paralog of *NR1F1*
*NPAS2*	Paralog of *CLOCK*; dimerization partner of *BMAL1/2*
*CSNK1E*	Phosphorylation of *PERs*
*PPAR-*γ	Metabolic regulator gene, Differentiation of adipocytes

Abbreviations: Circadian locomotor output cycles kaput (*CLOCK*), brain and muscle Arnt-like protein-1 (*BMAL1*), nuclear receptor subfamily 1, group D, member 1 (*REV-ERBα/REV-ERBβ*), nocturnin (*NOCT*), Peroxisome proliferator activated receptor gamma coactivator1 alpha (*PGC-1α*), period-circadian protein homolog 1/2/3 (*PER1/2/3*), cryptochrome 1 (*CRY1*), cryptochrome 2 (*CRY2*), arylhydrocarbon receptor nuclear translocator-like (*ARNTL*), timeless (*TIM*), retinoic acid-related orphan nuclear receptor (*ROR*), nuclear receptor subfamily 1, group F, member 1 (*NR1F1/RORα*), neuronal PAS domain protein 2 (*NPAS2/MOP4*), casein kinase 1 epsilon (*CSNK1E*), peroxisome proliferator-activated receptors gamma (*PPAR-γ*).

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
