# Peer review of "The Biological Clock of Liver Metabolism in Metabolic Dysfunction-Associated Steatohepatitis Progression to Hepatocellular Carcinoma"

_biomedicines, 2024, doi:10.3390/biomedicines12091961_

Round 1

Reviewer 1 Report (New Reviewer)

Comments and Suggestions for Authors

According to the authors "The present review provides an overview of the role of the liver's circadian rhythm in metabolic processes in health and disease". And this is completely true. The work will be interesting to a wide circle of medical specialists. The work is written in a professional and understandable way. Interesting illustrations are selected. I have only a few small comments for the text.

(1) The layout in Figure 1 should be improved. In the current version, the line numbers in the first column are confusing.

(2) Work style must be changed to Review. In the current version it is written Article. However, no investigation is carried out at work.

(3) The reference list must be presented in MDPI style. Each line of the current reference list  should be carefully reviewed.

Author Response

Reviewer 1

Thank you so much for your valuable comments and we appreciate your interest in our article.

  • The layout in Figure 1 should be improved. In the current version, the line numbers in the first column are confusing.

Figure 1 have been revised accordingly.

  • Work style must be changed to Review. In the current version it is written Article. However, no investigation is carried out at work.

Thank you for the valuable comments.

(3) The reference list must be presented in MDPI style. Each line of the current reference list should be carefully reviewed.

References have been revised accordingly.

Reviewer 2 Report (New Reviewer)

Comments and Suggestions for Authors

This review focuses on the study titled " The Biological Clock of Liver Metabolism in Metabolic Dysfunction-Associated Steatohepatitis Progression to Hepatocellular Carcinoma ". The document discusses the role of circadian rhythms in liver metabolism and their implications for metabolic dysfunction-associated steatohepatitis (MASH) and its progression to hepatocellular carcinoma (HCC). Circadian rhythms are endogenous cycles that regulate various biological processes, including metabolism and are influenced by clock genes that produce proteins with rhythmic patterns. The study highlights how disruptions in these rhythms can lead to metabolic disorders and cancer development in the liver. Understanding these mechanisms may provide insights into potential therapeutic strategies for treating liver diseases.

Comments:

1-     The authors need to expand on the pathophysiology of hepatocellular carcinoma (HCC) in the introduction section

2-     It would be helpful to mention the methods or approaches used to search articles related to the role of circadian rhythms in liver metabolism and their implications for MASH and its progression to HCC in the introduction.

3-     While the article discusses the association between circadian rhythms and liver diseases, it may not delve deeply enough into the specific molecular mechanisms involved. A more detailed exploration of how circadian disruptions lead to metabolic dysfunction could enhance understanding.

4-     The article might not adequately address other factors that can influence liver metabolism and disease, such as diet, lifestyle, and genetic predispositions. A comprehensive analysis should consider these variables to provide a clearer picture of the circadian impact.

5-     A list of abbreviations should be added to the manuscript

6-     To ensure the article is current and pioneering, it is recommended to update the literature cited. Keeping the references up-to-date will enhance the timeliness and relevance of the article's content.

7-     More Figures and Tables are needed to summarise the circadian genes and therapeutic targets in HCC

Author Response

Reviewer 2

Thank you so much for your valuable comments. We believe that the manuscript has been substantially improved because of your suggestions. We have addressed all the points suggested by the reviewer and revised the manuscript to reflect the changes. The changes are highlighted in the manuscript.

  • The authors need to expand on the pathophysiology of hepatocellular carcinoma (HCC) in the introduction section

      The manuscript has been revised accordingly (lines 99-109).

  • It would be helpful to mention the methods or approaches used to search articles related to the role of circadian rhythms in liver metabolism and their implications for MASH and its progression to HCC in the introduction.

The manuscript has been revised accordingly (lines 110-115).

  • While the article discusses the association between circadian rhythms and liver diseases, it may not delve deeply enough into the specific molecular mechanisms involved. A more detailed exploration of how circadian disruptions lead to metabolic dysfunction could enhance understanding.

      The manuscript has been revised accordingly (lines 468-487).

  • The article might not adequately address other factors that can influence liver metabolism and disease, such as diet, lifestyle, and genetic predispositions. A comprehensive analysis should consider these variables to provide a clearer picture of the circadian impact.

The present review article highlights the importance of the role of the liver’s circadian rhythm in metabolic processes in health and disease, emphasizing MASH progression and the oncogenic associations that lead to HCC. In this review, we have highlighted the pivotal role of apoptotic, epigenetic, and autophagic pathways in the regulation of the biological clock that leads to liver pathology. Hence, in the present review, we did not consider the influence of other factors such as diet, lifestyle, and genetic predispositions to make the review concise. We have added such purpose (lines 39 to 43).

  • A list of abbreviations should be added to the manuscript

     Abbreviations have been added accordingly.

  • To ensure the article is current and pioneering, it is recommended to update the literature cited. Keeping the references up to date will enhance the timeliness and relevance of the article's content.

     Recent references have been added accordingly.

  • More Figures and Tables are needed to summarize the circadian genes and therapeutic targets in HCC.

     As per the reviewer’s suggestion, a new figure (Figure 3) has been added to the manuscript.

Round 2

Reviewer 2 Report (New Reviewer)

Comments and Suggestions for Authors

The authors have addressed all of my concerns with the original manuscript

Author Response

Thank you so much.

This manuscript is a resubmission of an earlier submission. The following is a list of the peer review reports and author responses from that submission.

Round 1

Reviewer 1 Report

Comments and Suggestions for Authors

Pradeep and colleagues review the role of the liver circadian clock in health and disease with a special focus on the progression of MASH to HCC.

The usage of technical terms should be revisited in a number of cases. Some examples are:

-       chromosome organization (32) usually refers to epigenetics and higher order chromatin structure

-       cellular (I presume) rather than cell circadian/biological clock (39)

-       “output” genes (262) are actually output genes (without quotation marks) and the same is true for “core” clock genes (264).

Please explain abbreviations when they appear the first time, e.g. MASH, HCC (17), ESLD (38).

CLOCK, BMAL1 and NPAS2 are not regulated by CRYs and PERs but their transcriptional regulatory function is regulated by CRYs and PERs (53-62).

Please explain the paragraph about ATP1A1 which seems to be misplaced in this review article (305-321). No connection is made to the circadian clock, Table 1 does not list any common pathways or functions (321), and Figure 2 does not refer to circadian clock components. ATP1A1 shows even up in the conclusion while I could not find no reference to the “emerging studies … explored the link between the circadian clock and the signalosome” (465-466) in the manuscript.

Comments on the Quality of English Language

The English language could be improved throughout the manuscript. Often the language is not precise enough and statements are misleading. E.g. “The peripheral clocks play a major role in regulating food intake, …” (145). Food intake is a behavioral function controlled by the brain. Peripheral clocks might have indirect modifying functions in this process by regulating metabolic feedback signals but that is not what the following paragraph is about, which rather discusses nutrient processing functions. Etc…

Author Response

Review 1

  1. The usage of technical terms should be revisited in a number of cases. Some examples are:

-       chromosome organization (32) usually refers to epigenetics and higher order chromatin structure

-       cellular (I presume) rather than cell circadian/biological clock (39)

-       “output” genes (262) are actually output genes (without quotation marks) and the same is true for “core” clock genes (264).

Corrections have been made accordingly.

  1. Please explain abbreviations when they appear the first time, e.g. MASH, HCC (17), ESLD (38).

Abbreviations have been revised accordingly.

  1. CLOCK, BMAL1 and NPAS2 are not regulated by CRYs and PERs but their transcriptional regulatory function is regulated by CRYs and PERs (53-62).

Corrections have been made accordingly. (Lines:110-114)

  1. Please explain the paragraph about ATP1A1 which seems to be misplaced in this review article (305-321). No connection is made to the circadian clock, Table 1 does not list any common pathways or functions (321), and Figure 2 does not refer to circadian clock components. ATP1A1 shows even up in the conclusion while I could not find no reference to the “emerging studies … explored the link between the circadian clock and the signalosome” (465-466) in the manuscript.

As per the reviewer’s suggestion, we have removed the ATP1A1 paragraph.                                    

Table 1 lists the major circadian genes along with their function.

The manuscript has been revised to reflect the above changes.

  1. Comments on the Quality of English Language

The English language could be improved throughout the manuscript. Often the language is not precise enough and statements are misleading. E.g. “The peripheral clocks play a major role in regulating food

intake, …” (145). Food intake is a behavioral function controlled by the brain. Peripheral clocks might

have indirect modifying functions in this process by regulating metabolic feedback signals but that is not what the following paragraph is about, which rather discusses nutrient processing functions. Etc…

The manuscript has been revised accordingly

Reviewer 2 Report

Comments and Suggestions for Authors

I really enjoyed reviewing this paper!

The review is well written and widely comprehensive, the figures are nice and useful to the reader.

I would add some clinical implications of these concepts, for example mentioning the therapies to prevent MASH progression to HCC (in this regard cite the recent MA: PMID: 33721336 )

Author Response

Review 2

English language fine. No issues detected

I really enjoyed reviewing this paper!

The review is well written and widely comprehensive, the figures are nice and useful to the reader.

Thank you for the valuable comments.

I would add some clinical implications of these concepts, for example mentioning the therapies to prevent MASH progression to HCC (in this regard cite the recent MA: PMID: 33721336 )

A description has been added about the therapies of HCC under the section “Circadian     genes and therapeutic targets in HCC”.(Lines:461-463)

Reviewer 3 Report

Comments and Suggestions for Authors

Biomedicines-2762576

Type of manuscript: Review

Title: The Biological Clock of Liver Metabolism in MASH Progression to Hepatocellular Carcinoma

Authors: Pradeep KR, Udoh U, Finley R, Sodhi K, Pierre S and Sanabria J

This paper is a review paper; however, on Line 1 of the provided PDF file, it is labeled as an 'Article.' Furthermore, based on my recent experience publishing a review paper in ‘Biomedicines’, it can be inferred that this review article has various significant shortcomings. Considering the following points, it is evident that further improvements are necessary.

[Major Concerns]

1.    I searched for review papers related to the key words of this paper on PubMed and found about a dozen. However, a significant number of these review papers were not cited as references. Generally, when writing a review paper, it is considered appropriate to review topics that have not been covered in existing review papers or to discuss more advanced research results. However, the authors give the impression that they are reviewing the field for the first time, which I believe is not appropriate.

2.    When discussing a particular disease, it is essential to mention the latest epidemiology or statistical data related to these conditions. Among the diseases described in this review paper, MASH and HCC are prominent; however, there is no mention of global statistical data for these conditions. Especially regarding HCC, the statement in Lines 374-375 simply reads, 'Hepatocellular carcinoma (HCC), the most common lethal form of liver cancer, is one of the leading causes of cancer-related deaths worldwide.' It is crucial to appropriately cite recent statistical data and relevant references for a comprehensive understanding.

3.    Abbreviations: Most journals require that an abbreviation be spelled out at its first occurrence in the text, followed by the abbreviation in parentheses. (Exception: If the abbreviation is on the journal's list of permitted abbreviations, this need not be done.) Thereafter, only the abbreviation may be used. Note also that abbreviations need to be independently defined in the abstract and the main text of the paper. Abbreviations need not be introduced if they are not used again.

4.    Some abbreviations have been mentioned earlier, but they are being repeated later. Therefore, it would be beneficial to systematically review and clarify the abbreviation usage from the beginning. In the final part of the paper, the authors have compiled several abbreviations; however, it is impractical to list all the mentioned abbreviations in the paper with this summary. Therefore, the authors suggest that, when necessary, defining abbreviations first and then consistently using them from the beginning to the end of the paper would be more appropriate.

5.    In cases where abbreviations are used within figures or tables, please list these abbreviations along with their corresponding full names in the figure legends or at the bottom of corresponding tables. If there are two or more abbreviations, arrange them in alphabetical order.

6.    English: The English composition of the paper is generally well-done. However, some names of disease or compound names are written in uppercase letters even though they are not the first letter of the sentence or proper nouns. Please make corrections throughout the text and in the figures.

7.    Naming of proteins and genes: When composing a review paper, it is inevitable to mention numerous protein and corresponding gene names. Despite the regulation specifying the italicization of gene names, it is not being adhered to at all. This has led to considerable confusion. Therefore, please rectify and clarify the notation of protein and gene names in accordance with the prescribed italicization for gene names. Furthermore, since the notation for specific proteins or gene names is inconsistent, please verify all of them again. Examples: BCL-2 at Line 198 vs. Bcl-2 at Line 421; Na/K-ATPase at Line 308 vs. Na+/K+-ATPase at Line 318; etc.

8.    Citation of references and lists of references: The examination of the submitted manuscript's PDF gives the impression that it has been formatted according to the submission guidelines of a different journal. Would it be common sense to adjust the manuscript format to match the submission guidelines of the journal to which it is being submitted? Additionally, citation reference 1 should appear first in the Introduction section, but it is currently missing. Please make the necessary revisions to the references, aligning them with the formatting guidelines of the ‘Biomedicines’ journal.

[Minor Concerns]

1.    Line 17: Define MASH and HCC at the Abstract.

2.    Line 38: Define ESLD.

3.    Line 114: ‘BMAL1-/-’ should be written as ‘BMAL1-/-’.

4.    Line 158: ‘24h-rhythm’: The notation used here seems inappropriate, so please rewrite it.

5.    Line 257: ‘Histone’ should be written as ‘histone’.

6.    Line 371: For the Figure 2, it is needed reference(s).

7.    Line 374: HCC has already been abbreviated. Just use HCC here.

8.    Line 428: Define TNM.

9.    Line 453: Define lncRNAs.

10.    Reference section: Author should consult and peruse carefully recent issues of the journal, Biomedicines, for format and style. Also double-check the abbreviations of journal names

11.    Several references are missing page information. Examples: 23, 49, 93, 98, 113, 115, 120, 123, 125, etc.

Overall, the manuscript can be considered to publication after major revision as indicated above.

Comments on the Quality of English Language

Biomedicines-2762576

Type of manuscript: Review

Title: The Biological Clock of Liver Metabolism in MASH Progression to Hepatocellular Carcinoma

Authors: Pradeep KR, Udoh U, Finley R, Sodhi K, Pierre S and Sanabria J

This paper is a review paper; however, on Line 1 of the provided PDF file, it is labeled as an 'Article.' Furthermore, based on my recent experience publishing a review paper in ‘Biomedicines’, it can be inferred that this review article has various significant shortcomings. Considering the following points, it is evident that further improvements are necessary.

[Major Concerns]

1.    I searched for review papers related to the key words of this paper on PubMed and found about a dozen. However, a significant number of these review papers were not cited as references. Generally, when writing a review paper, it is considered appropriate to review topics that have not been covered in existing review papers or to discuss more advanced research results. However, the authors give the impression that they are reviewing the field for the first time, which I believe is not appropriate.

2.    When discussing a particular disease, it is essential to mention the latest epidemiology or statistical data related to these conditions. Among the diseases described in this review paper, MASH and HCC are prominent; however, there is no mention of global statistical data for these conditions. Especially regarding HCC, the statement in Lines 374-375 simply reads, 'Hepatocellular carcinoma (HCC), the most common lethal form of liver cancer, is one of the leading causes of cancer-related deaths worldwide.' It is crucial to appropriately cite recent statistical data and relevant references for a comprehensive understanding.

3.    Abbreviations: Most journals require that an abbreviation be spelled out at its first occurrence in the text, followed by the abbreviation in parentheses. (Exception: If the abbreviation is on the journal's list of permitted abbreviations, this need not be done.) Thereafter, only the abbreviation may be used. Note also that abbreviations need to be independently defined in the abstract and the main text of the paper. Abbreviations need not be introduced if they are not used again.

4.    Some abbreviations have been mentioned earlier, but they are being repeated later. Therefore, it would be beneficial to systematically review and clarify the abbreviation usage from the beginning. In the final part of the paper, the authors have compiled several abbreviations; however, it is impractical to list all the mentioned abbreviations in the paper with this summary. Therefore, the authors suggest that, when necessary, defining abbreviations first and then consistently using them from the beginning to the end of the paper would be more appropriate.

5.    In cases where abbreviations are used within figures or tables, please list these abbreviations along with their corresponding full names in the figure legends or at the bottom of corresponding tables. If there are two or more abbreviations, arrange them in alphabetical order.

6.    English: The English composition of the paper is generally well-done. However, some names of disease or compound names are written in uppercase letters even though they are not the first letter of the sentence or proper nouns. Please make corrections throughout the text and in the figures.

7.    Naming of proteins and genes: When composing a review paper, it is inevitable to mention numerous protein and corresponding gene names. Despite the regulation specifying the italicization of gene names, it is not being adhered to at all. This has led to considerable confusion. Therefore, please rectify and clarify the notation of protein and gene names in accordance with the prescribed italicization for gene names. Furthermore, since the notation for specific proteins or gene names is inconsistent, please verify all of them again. Examples: BCL-2 at Line 198 vs. Bcl-2 at Line 421; Na/K-ATPase at Line 308 vs. Na+/K+-ATPase at Line 318; etc.

8.    Citation of references and lists of references: The examination of the submitted manuscript's PDF gives the impression that it has been formatted according to the submission guidelines of a different journal. Would it be common sense to adjust the manuscript format to match the submission guidelines of the journal to which it is being submitted? Additionally, citation reference 1 should appear first in the Introduction section, but it is currently missing. Please make the necessary revisions to the references, aligning them with the formatting guidelines of the ‘Biomedicines’ journal.

[Minor Concerns]

1.    Line 17: Define MASH and HCC at the Abstract.

2.    Line 38: Define ESLD.

3.    Line 114: ‘BMAL1-/-’ should be written as ‘BMAL1-/-’.

4.    Line 158: ‘24h-rhythm’: The notation used here seems inappropriate, so please rewrite it.

5.    Line 257: ‘Histone’ should be written as ‘histone’.

6.    Line 371: For the Figure 2, it is needed reference(s).

7.    Line 374: HCC has already been abbreviated. Just use HCC here.

8.    Line 428: Define TNM.

9.    Line 453: Define lncRNAs.

10.    Reference section: Author should consult and peruse carefully recent issues of the journal, Biomedicines, for format and style. Also double-check the abbreviations of journal names

11.    Several references are missing page information. Examples: 23, 49, 93, 98, 113, 115, 120, 123, 125, etc.

Overall, the manuscript can be considered to publication after major revision as indicated above.

Author Response

Review 3

This paper is a review paper; however, on Line 1 of the provided PDF file, it is labeled as an 'Article.' Furthermore, based on my recent experience publishing a review paper in ‘Biomedicines’, it can be inferred that this review article has various significant shortcomings. Considering the following points, it is evident that further improvements are necessary.

Thank you for the valuable comments.

[Major Concerns]

  1. I searched for review papers related to the key words of this paper on PubMed and found about a dozen. However, a significant number of these review papers were not cited as references. Generally, when writing a review paper, it is considered appropriate to review topics that have not been covered in existing review papers or to discuss more advanced research results. However, the authors give the impression that they are reviewing the field for the first time, which I believe is not appropriate.

The present review article highlights the importance of the role of the liver’s circadian rhythm in metabolic processes in health and disease, emphasizing MASH progression and the oncogenic associations that lead to HCC. In this review, we have highlighted the pivotal role of apoptotic, epigenetic, and autophagic pathways in the regulation of the biological clock that leads to liver pathology. Based on the reviewer’s comment, revisions have been made to more clearly present the novelty of this manuscript in the context of previous literature.  We are also citing additional recent references. (Lines:135-137;171-175;386-395).

  1. When discussing a particular disease, it is essential to mention the latest epidemiology or statistical data related to these conditions. Among the diseases described in this review paper, MASH and HCC are prominent; however, there is no mention of global statistical data for these conditions. Especially regarding HCC, the statement in Lines 374-375 simply reads, 'Hepatocellular carcinoma (HCC), the most common lethal form of liver cancer, is one of the leading causes of cancer-related deaths worldwide.' It is crucial to appropriately cite recent statistical data and relevant references for a comprehensive understanding.

The recent statistical data of HCC has been included along with relevant references (Lines;386-395).

  1. Abbreviations: Most journals require that an abbreviation be spelled out at its first occurrence in the text, followed by the abbreviation in parentheses. (Exception: If the abbreviation is on the journal's list of permitted abbreviations, this need not be done.) Thereafter, only the abbreviation may be used. Note also that abbreviations need to be independently defined in the abstract and the main text of the paper. Abbreviations need not be introduced if they are not used again.

Abbreviations have been revised accordingly.

  1. Some abbreviations have been mentioned earlier, but they are being repeated later. Therefore, it would be beneficial to systematically review and clarify the abbreviation usage from the beginning. In the final part of the paper, the authors have compiled several abbreviations; however, it is impractical to list all the mentioned abbreviations in the paper with this summary. Therefore, the authors suggest that, when necessary, defining abbreviations first and then consistently using them from the beginning to the end of the paper would be more appropriate.

Abbreviations have been revised accordingly.

  1. In cases where abbreviations are used within figures or tables, please list these abbreviations along with their corresponding full names in the figure legends or at the bottom of corresponding tables. If there are two or more abbreviations, arrange them in alphabetical order.

Abbreviations have been revised accordingly.

  1. English: The English composition of the paper is generally well-done. However, some names of disease or compound names are written in uppercase letters even though they are not the first letter of the sentence or proper nouns. Please make corrections throughout the text and in the figures.

The manuscript has been revised accordingly.

  1. Naming of proteins and genes: When composing a review paper, it is inevitable to mention numerous protein and corresponding gene names. Despite the regulation specifying the italicization of gene names, it is not being adhered to at all. This has led to considerable confusion. Therefore, please rectify and clarify the notation of protein and gene names in accordance with the prescribed italicization for gene names. Furthermore, since the notation for specific proteins or gene names is inconsistent, please verify all of them again. Examples: BCL-2 at Line 198 vs. Bcl-2 at Line 421; Na/K-ATPase at Line 308 vs. Na+/K+-ATPase at Line 318; etc.

The manuscript has been revised accordingly.

  1. Citation of references and lists of references: The examination of the submitted manuscript's PDF gives the impression that it has been formatted according to the submission guidelines of a different journal. Would it be common sense to adjust the manuscript format to match the submission guidelines of the journal to which it is being submitted? Additionally, citation reference 1 should appear first in the Introduction section, but it is currently missing. Please make the necessary revisions to the references, aligning them with the formatting guidelines of the ‘Biomedicines’ journal.

The submission of the initial draft of the manuscript for ‘Biomedicines’ follows “Free-format submission”. Further formatting of the manuscript and the references according to the journal to generate the peer review PDF was done by the Editorial team.

[Minor Concerns]

  1. Line 17: Define MASH and HCC at the Abstract.

  1. Line 38: Define ESLD.

  1. Line 114: ‘BMAL1-/-’ should be written as ‘BMAL1-/-’.

  1. Line 158: ‘24h-rhythm’: The notation used here seems inappropriate, so please rewrite it.

  1. Line 257: ‘Histone’ should be written as ‘histone’.

  1. Line 371: For the Figure 2, it is needed reference(s).
  2. Line 374: HCC has already been abbreviated. Just use HCC here.

  1. Line 428: Define TNM.

  1. Line 453: Define lncRNAs.

  1. Reference section: Author should consult and peruse carefully recent issues of the journal, Biomedicines, for format and style. Also double-check the abbreviations of journal names

  1. Several references are missing page information. Examples: 23, 49, 93, 98, 113, 115, 120, 123, 125, etc.